https://doi.org/10.1038/s42003-022-03165-7　　**OPEN**
# Transgenic expression of *Nix* converts genetic females into males and allows automated sex sorting in *Aedes albopictus*

Célia Lutrat [1,2,3,4✉], Roenick P. Olmo [4], Thierry Baldet [1,2,5], Jérémy Bouyer [1,6,7] & Eric Marois [4,7✉]

*Aedes albopictus* is a major vector of arboviruses. Better understanding of its sex determination is crucial for developing mosquito control tools, especially genetic sexing strains. In *Aedes aegypti*, *Nix* is the primary gene responsible for masculinization and *Nix*-expressing genetic females develop into fertile, albeit flightless, males. In *Ae. albopictus*, *Nix* has also been implicated in masculinization but its role remains to be further characterized. In this work, we establish *Ae. albopictus* transgenic lines ectopically expressing *Nix*. Several are composed exclusively of genetic females, with transgenic individuals being phenotypic and functional males due to the expression of the *Nix* transgene. Their reproductive fitness is marginally impaired, while their flight performance is similar to controls. Overall, our results show that *Nix* is sufficient for full masculinization in *Ae. albopictus*. Moreover, the transgene construct contains a fluorescence marker allowing efficient automated sex sorting. Consequently, such strains constitute valuable sexing strains for genetic control.

[1] CIRAD, UMR ASTRE, F-34398 Montpellier, France. [2] ASTRE, CIRAD, INRA, Univ. Montpellier, Montpellier, France. [3] Université de Montpellier, Montpellier, France. [4] CNRS UPR9022, INSERM U1257, Université de Strasbourg, F-67084 Strasbourg, France. [5] CIRAD, UMR ASTRE, F-97490 Sainte-Clotilde, Réunion. [6] CIRAD, UMR ASTRE, F-97410 Saint-Pierre, Réunion. [7] These authors jointly supervised this work: Jérémy Bouyer, Eric Marois. ✉email: celia.lutrat@outlook.com; e.marois@unistra.fr

Aedes albopictus can transmit the main human arboviruses including dengue, chikungunya, yellow fever and Zika viruses[1–6]. Given the global expansion of this mosquito[7,8] and its increasing insecticide resistance[9,10], new and more sustainable vector control tools are urgently needed[11]. Genetic control methods to suppress mosquito population, including the Sterile Insect Technique[12] and the Incompatible Insect Technique[13], have proven effective in several field trials[14–17]. These methods rely on mass releases of male mosquitoes that compete with their field counterparts for mating. However, in order to upscale the mass production process, an automated and more cost-efficient method for separating males from females is still necessary[18,19]. Unfortunately, reliable sexing strains are not yet available on an operational scale in Ae. albopictus, in part due to the limited knowledge of the sex determination in this species.

Insects exhibit a large diversity of primary signals that activate the sex determination cascade. These signals are poorly conserved between insect genera and few have been discovered so far. In Drosophila melanogaster, the X chromosome:autosome ratio is the primary signal for sex determination and dosage compensation. While the Y chromosome is essential for male fertility, it does not determine phenotypic sex[20]. In some non-Drosophilid Diptera, the primary signal relies on a male factor (M-factor) located either on the Y chromosome or in the M-locus of an autosome. Aedes mosquitoes carry an M-factor located within an M-locus on the first pair of autosomes, which is called Nix[21–23]. Nix was first identified in Ae. aegypti, where it consists of two exons separated by a 99-kb intron and encodes a 288-amino acid protein[21]. Stable transgenic expression of Nix in Ae. aegypti has been shown to be sufficient for converting all genetic females into phenotypic fertile males, confirming its role[24]. However, these phenotypic males were unable to fly, most likely because they lacked myo-sex, another gene closely linked to the M-locus encoding a male-specific flight muscle myosin. In Ae. albopictus, Nix is also located on chromosome I and comprises two exons with high similarity to Ae. aegypti Nix and a much shorter intron[25], as well as two additional exons uncovered recently[22]. Nix disruption using CRISPR/Cas9 leads to partial feminization of males, confirming the role of Nix as the M-factor in this species[22].

It remains to be shown whether Ae. albopictus Nix alone is sufficient for masculinization and if genetic females transformed into phenotypic males by transgenic expression of Nix would be fertile and able to fly.

In this work, we generated transgenic Ae. albopictus mosquitoes expressing the four main Nix isoforms in genetic females and showed that all can induce partial to complete masculinization. Some of these lines constitute valuable genetic sexing strains, allowing high-throughput sex sorting at the neonate stage.

## Results

**Obtaining Nix-expressing transgenic Ae. albopictus lines.** Because additional exons were unknown when this project began, we first built a transgenesis plasmid comprising only Nix exon 1, intron 1 and exon 2 under the control of the endogenous Nix promoter (~2 kb preceding the ATG start codon, see Supplementary Note 1). We included an OpIE2-eGFP fluorescence marker for identification of transgenic individuals (Fig. 1c). About 300 Ae. albopictus embryos were injected with this plasmid. Among the surviving G0 adults, we obtained 24 males showing transient expression (TE) of the fluorescent marker and few females, some of which showing deformed genitalia at the pupal stage. After crossing en masse TE males with wild-types females, we recovered 29 eGFP positive G1 pupae, including 22 males and 7 females. We further outcrossed the 22 positive males

en masse with wild-type females. In the G2 generation, we obtained 395 males expressing GFP while 53 were negative, and 185 females expressing eGFP while 151 were negative. Of these, 12 randomly selected eGFP males were outcrossed individually with WT females, creating the lines named herein SM1 to SM12. Two crosses, (SM8 and SM10), did not produce eggs and were not further investigated. In all the other progenies but SM4, 100% of the G3 males expressed eGFP. GFP positive females were also observed, although these were rare or absent in most lines. Interestingly, in SM9 57% of eGFP positive females (larger body size and female antennae) showed an intersex phenotype with deformed genitalia (Supplementary Fig. 1). Transgenic lines were amplified over several generations and in five of them (SM1, SM3, SM7, SM9, and SM12), 100% of the males were GFP positive.

The second injection mix comprised three different transgenesis plasmids to express the newly reported Nix isoforms[22], labelled with different fluorescence markers (Fig. 1a, b and d with DsRed, YFP and eGFP respectively) under the control of the Ae. aegypti poly-ubiquitin promoter (PUb). We injected about 700 embryos with this mix, from which approximately 100 TE individuals were obtained. Similar to our previous observations, we obtained mostly male G0 TE individuals that were split in five pools of males and outcrossed en masse with wild-type females. Strikingly, in the G1 generation, 70–100% of the transgenic individuals of each pool were males. Some of the transgenic females in at least three pools were deformed or partially masculinised. Most transgenic G1 individuals expressed more than one fluorescent marker, indicating that they carried multiple piggyBac insertions with different transgenes. In each of five G1 pools, 8 transgenic males were randomly selected and outcrossed individually with WT females in order to generate lines carrying a single fluorescent marker. From these 40 individual founder males, we retained 7 independent lines in which 100% of the males were fluorescent (namely 1.2R, 1.2G, 2.2G, 3.1YR, 3.1G, 4.4Y, 5.1GR) for further analysis.

**Identification of transgenic lines devoid of an M-locus.** During characterization of Nix-expressing transgenic lines, we observed strong genetic linkage between fluorescence and the male phenotype. This could be due either to transgenic fluorescent males actually being masculinised genetic females, resulting in mosquito lines lacking natural males; or to the insertion of the piggyBac plasmid cassette within or near the endogenous M-locus, resulting in genetic linkage between the fluorescent transgene and male sex. To distinguish between these two possibilities, we tested lines with 100% fluorescent males for the presence of endogenous and exogenous Nix by PCR (e.g. see Supplementary Fig. 2). In eight out of twelve lines, endogenous Nix was detected in the GFP positive transgenic males. Therefore, in these lines, maleness was natural and at least one copy of the transgene was M-linked. In contrast, four of our mosquito lines were devoid of endogenous Nix, namely SM9, 1.2G, 2.2G and 3.1G. Interestingly, these four lines possessed the GFP fluorescent marker, thus, they expressed the shortest Nix isoforms encoded by exons 1 and 2 only. In some lines marked by YFP and/or DsRed fluorescence, a fraction of the males lacked the M-locus, while others possessed it. However, the M-deprived males did not sire any progeny, thus they were possibly sterile.

Single males from the four M-free GFP lines (hereafter termed pseudo-males) were backcrossed to wild-type females for several generations to eliminate additional non-fully masculinizing transgene insertions. Following this step, we obtained three lines (SM9, 1.2G and 3.1G) where only pseudo-males showed eGFP fluorescence, without residual fluorescent females or intersex individuals. Further tests and analyses were performed on the

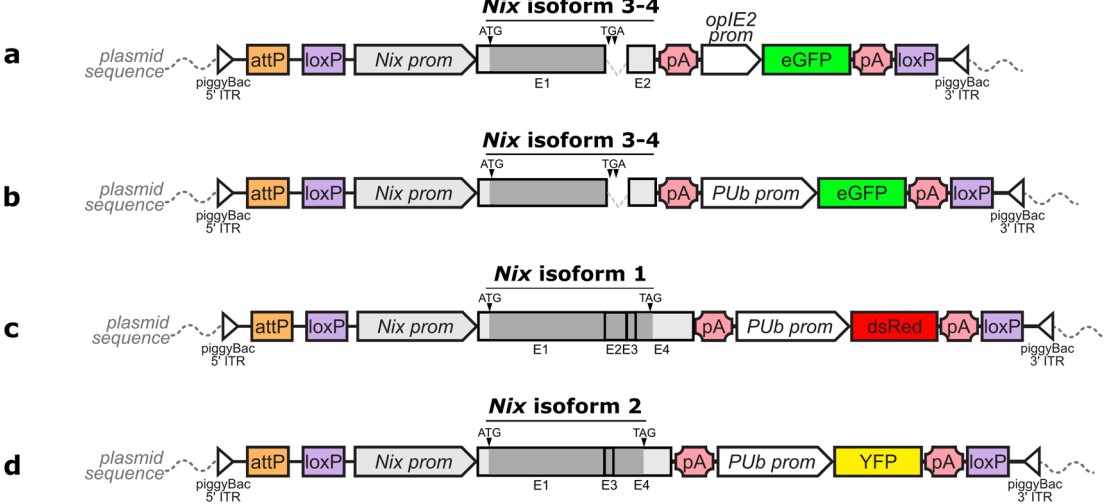

**Fig. 1 Schematic representation of *Ae. albopictus* Nix isoform cloning in the injected plasmids.** Isoform naming follows work published in ref. [22]. Grey boxes are *Nix* exons, with darker grey being the translated parts. White triangles are *piggyBac* 5′ and 3′ inverted terminal repeat (ITR) sequences, orange boxes represent an AttP landing site included for potential future purposes, purple boxes represent loxP recombination sites, white arrows are promoters, pink polygons are SV40 polyA sequences, green, red and yellow boxes are *eGFP*, *DsRed* and *YFP* gene sequences, respectively. Drawing not to scale. Plasmids carry **a** *Nix* isoforms 3–4 and a *GFP* marker gene under the control of the *OpIE2* promoter **b** *Nix* isoforms 3–4 and a *GFP* marker gene under the control of the polyubiquitin promoter, **c** *Nix* isoform 1 and a *DsRed* marker gene under the control of the polyubiquitin promoter, and **d** *Nix* isoform 2 and a *YFP* marker gene under the control of the polyubiquitin promoter. Detailed plasmid sequences can be found under Addgene references #173505, #173665, #173666, #173667.

SM9 line, and some experiments were replicated on the other two lines.

**Confirmation of the role of *Nix* transgenes in masculinization.** To further confirm the role of *Nix* transgenes in masculinization, we injected SM9 embryos with a plasmid expressing CRE recombinase in order to excise the *Nix-eGFP* transgenic cassette, which is flanked by lox recombination sites (Fig. 1). At the pupal stage, efficiently injected individuals showed a striking loss of eGFP expression at the injection site (posterior pole), indicative of lox cassette excision. While these pupae should have developed into phenotypic males (due to the eGFP-marked *Nix* transgene cassette), they showed female genitalia, hence demasculinization of their posterior pole (Fig. 2a–c, Supplementary Fig. 3). Adult mosquitoes hatching from these pupae showed antero-posterior gynandromorphism, having male heads and female genitalia. Strikingly, a single individual, which arose from an embryo accidentally injected in the anterior pole rather than in the posterior, showed the opposite gynandromorphic phenotype, with a female head and male genitalia (Fig. 2d). These results confirmed that maleness in the SM9 line results from the transgene's activity, which can be abolished by CRE/lox excision. The co-existence of male and female tissues in the same individual also illustrates that sex determination is tissue-autonomous in *Ae. albopictus*.

**Characterization of *Nix*-expressing pseudo-males.** To evaluate whether the *Nix-eGFP* cassette was stable and adult pseudo-males fully viable, we first determined the sex ratio in comparison to that of the parental wild-type (WT) line (Fig. 3a, Supplementary Table 2). Male ratios from SM9 (sex-ratio estimate ± SE = 0.56 ± 0.05), 1.2G (0.49 ± 0.05) and 3.1G (0.54 ± 0.05) transgenic strains were not significantly different from that of the WT line (0.52 ± 0.09, SM9 vs. WT $p$-value = 0.165, 1.2G vs. WT $p$-value = 0.528, 3.1G vs. WT $p$-value = 0.760).

In *Aedes* mosquitoes, males and females display a significant size dimorphism, with females having a larger body size[26]. We determined the body size of pseudo-males using wing length as a proxy[27]. Our results showed no significant difference between the size of the wild-type and transgenic males (WT male vs. SM9 pseudo-male $p$ = 0.998), while both were significantly different from females (WT male vs. WT female $p$-value < 0.001, SM9 pseudo-male vs. WT female $p$-value < 0.001, Fig. 3b, Supplementary Data 2).

*Nix* expression levels in pseudo-males vs. wild-type males were compared by RT-qPCR at the pupal stage (Fig. 4a, Supplementary Data 3). Interestingly, pseudo-males from three different lines expressed *Nix* at a similar level to wild-type males (SM9 males vs. WT males $p$-value = 0.501, 1.2G males vs. WT males $p$-value = 0.391, 3.1G males vs. WT males $p$-value = 0.626). Thus, we inquired if the downstream double switch genes, *doublesex* (*dsx*) and *fruitless* (*fru*), showed male-specific splicing products in pseudo-males. For this, we performed RT-PCR on pseudo-males of four transgenic lines. Results revealed that pseudo-males displayed the same splicing pattern as WT males for both genes (Supplementary Fig. 4).

**Expression of *myo-sex* orthologues in *Ae. albopictus* pseudo-males.** Contrarily to *Ae. aegypti* where the lack of the M-linked gene *myo-sex* resulted in flightless pseudo-males[24], our *Ae. albopictus* pseudo-males were readily able to fly. Consequently, we wondered whether an essential orthologue of *myo-sex* was present in the *Ae. albopictus*'s M-locus. Using the current version of the *Ae. albopictus* genome (Vectorbase release 52, 20 May 2021), we found two homologous copies of *Ae. aegypti myo-sex* located on two distinct unplaced scaffolds (SWKY01000423 and SWKY01000369), annotated under the identifier references LOC109430926 and LOC109412105, respectively[24,28]. We designed primer pairs specific to each of these two genes, exploiting differences in their non-coding regions. Interestingly, one of these PCR markers suggested the presence of a male-specific copy of *myo-sex*, characterized by a 664 bp deletion in its non-coding sequence (Supplementary Note 2). These results do not align with the currently available genomic data, and suggest that a third, M-linked copy of the *myo-sex* gene exists or that the

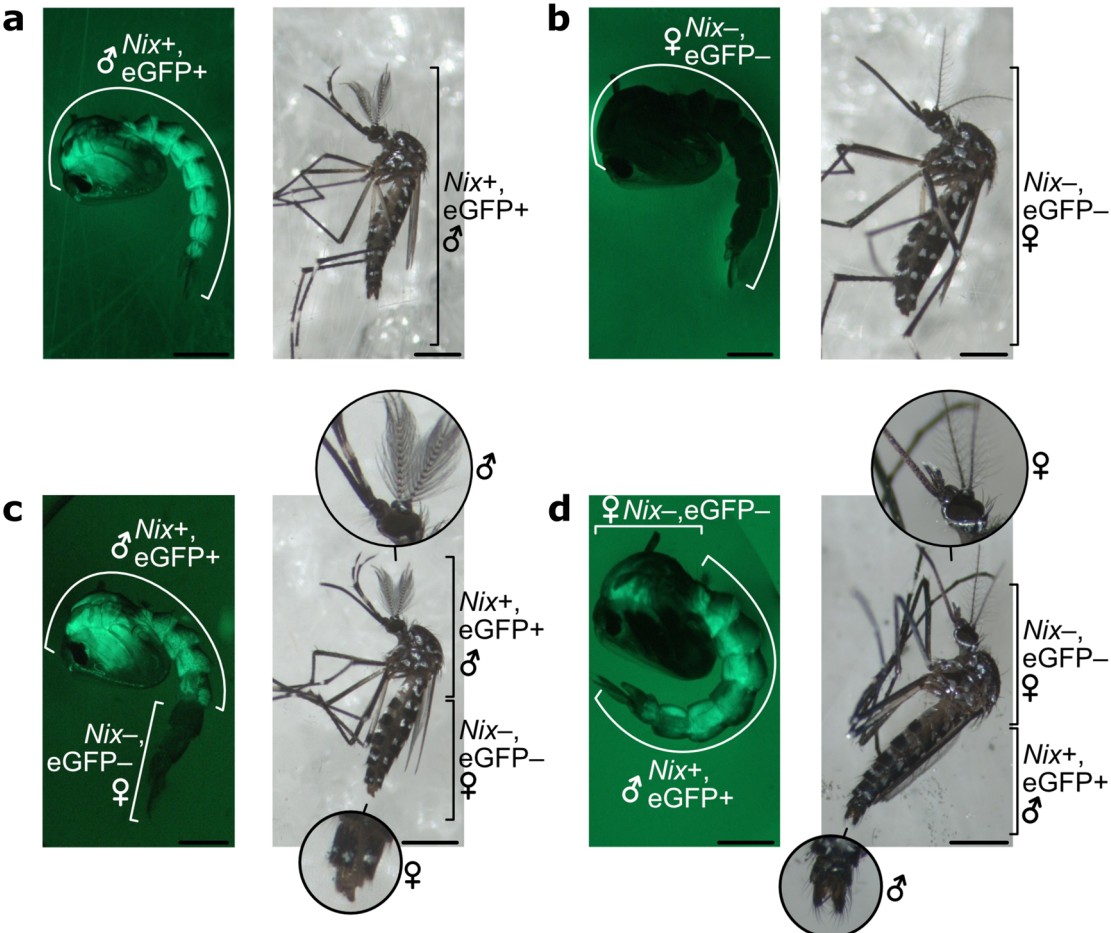

**Fig. 2 Tissue demasculinization upon CRE/lox excision of the transgenic *Nix* cassette. a** Representative transgenic male pupa and male adult from the SM9 line. **b** Representative non-transgenic female pupa and female adult from the SM9 line. **c** Transgenic SM9 male pupa and adult injected as embryos in the posterior pole with CRE-recombinase that excised the Nix-eGFP cassette in the injected region. These individuals show a male anterior body with female genitalia. **d** Transgenic SM9 male pupa and adult injected with CRE-recombinase in the anterior pole of the embryo. Note the female anterior body and male genitalia. Scale bars in the bottom right corner of each picture represent 1 mm.

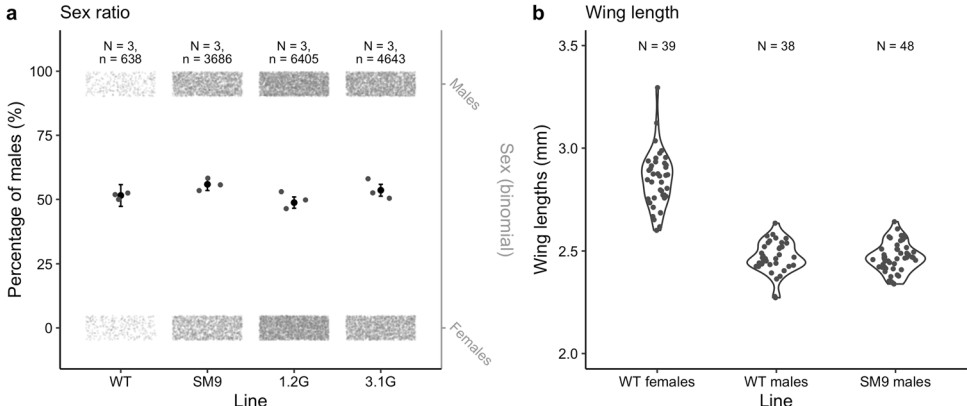

**Fig. 3 Comparison of sex ratios and wing lengths between transgenic and WT lines. a** Sex ratio comparison between the WT line and the SM9, 1.2G and 3.1G transgenic lines. Sex ratio of the WT line was counted manually on $N = 3$ independent batches of pupae, while sex ratios of transgenic lines were counted on $N = 3$ independent batches of neonate larvae using COPAS. Grey dots in rectangles represent the total numbers of males and females (right y-axis). Grey dots in the middle part represent the sex ratio of each replicate. Black dots are the estimate values with vertical lines being 95% confidence intervals. Sex ratios were compared using linear generalised mixed-effect model. None of the sex ratios were significantly different from that of the WT line: SM9 vs. WT *p*-value = 0.165, 1.2G vs. WT *p*-value = 0.528, 3.1G vs. WT *p*-value = 0.760. **b** Wing length comparison between $N = 38$ wild-type males, $N = 39$ wild-type females and $N = 48$ *Nix*-expressing SM9 pseudo-males represented as violin plots with jitter grey data points. Wing length was measured on ImageJ software from pictures of dissected right wings taken under a binocular microscope. Comparisons were performed using linear model: WT male vs. WT female *p*-value < 0.001, SM9 male vs. WT female *p*-value < 0.001, WT male vs. SM9 male *p*-value = 0.998.

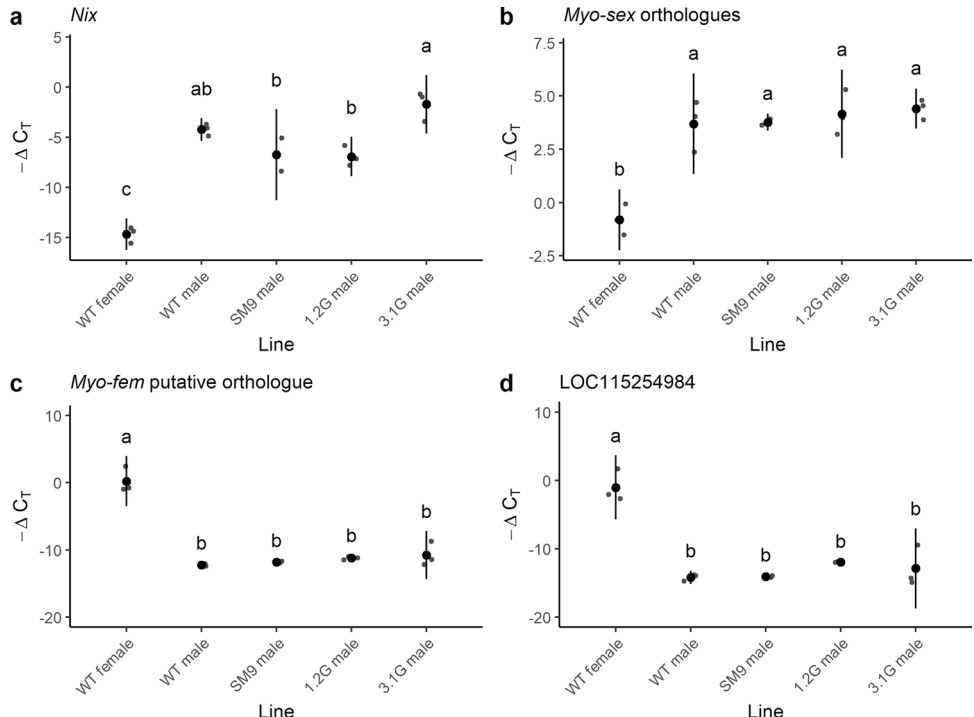

**Fig. 4 Relative expression of *Nix*, *myo-sex* and *myo-fem* orthologue in *Ae. albopictus* transgenic males.** RT-qPCR results are represented by $-\Delta C_T$, which reflects the relative expression level of each gene in a given treatment, $C_T$ values being inversely proportional to the expression levels. *AalRpS7* was used as endogenous reference gene. Grey dots represent each data point. Black dots represent the mean value of the $N = 3$ biological replicates, vertical lines represent 95% confidence intervals. On each panel, distinct letters represent significant difference in an ANOVA followed by a pairwise Tukey test ($p$-value < 0.001). **a** *Nix* relative expression. **b** Relative comparison of *myo-sex* orthologues total expression levels. **c** Relative expression of the candidate orthologue of the *Ae. aegypti myo-fem* gene, LOC109402113. **d** Relative expression of the candidate orthologue LOC115254984, which is annotated as a putative pseudo-gene. This primer pair could also amplify LOC115254986 due to high sequence similarity but this other pseudo-gene seems not to be expressed.

genome assembly concerning one of the above-mentioned copies is erroneous possibly due to high similarity between them. Hence, we compared the expression levels of the *Ae. albopictus* orthologues of *myo-sex* in *Nix*-expressing pseudo-males lacking the M-linked copy vs. WT males by RT-qPCR. All *myo-sex* copies being 100% identical at the cDNA level, RT-qPCR reflects global expression of all copies combined. Our results showed that pseudo-males expressed *myo-sex* at a level similar to wild-type males (SM9 males vs. WT males $p$-value = 1.000, 1.2G males vs. WT males $p$-value = 0.960, 3.1G males vs. WT males $p$-value = 0.843), and that this level was approximately 20 fold higher compared to WT females ($p$-values < 0.001 for all combinations, Fig. 4b, Supplementary Data 3). These results suggest that one or several non M-linked, endogenous *myo-sex*-like copies are efficiently upregulated in pseudo-males.

**Expression of *myo-fem* orthologue in *Ae. albopictus* pseudo-males.** Another sex-specifically expressed flight gene is *myo-fem*, described in *Ae. aegypti* as essential for female flight[29]. In *Ae. albopictus*, this gene has several potential orthologues. We compared their expression by RT-PCR between WT males and females and identified a putative *myo-fem* orthologue (LOC109402113) based on its strong female-specific expression pattern. RT-qPCR results revealed that pseudo-males express *myo-fem* at similarly low levels as wild-type males (SM9 males vs. WT males $p$-value = 0.995, 1.2G males vs. WT males $p$-value = 0.841, 3.1G males vs. WT males $p$-value = 0.614, Fig. 4c, Supplementary Data 3). Gene expression in all males tested was approximately 10,000 fold lower than in wild-type females ($p$-values < 0.001 for all combinations). Notably, another potential

orthologue located on the same genomic scaffold as LOC109402113 and annotated as a pseudo-gene with detectable RNA expression, LOC115254984, displayed a very similar expression profile (Fig. 4d, Supplementary Data 3).

**Flight ability and reproductive fitness of SM9 pseudo-males.** Since we observed that genes potentially involved in flight are regulated similarly in *Nix*-expressing pseudo-males comparing to wild-type counterparts, we tested the SM9 pseudo-males' flight ability by performing a flight test as described in ref. [30]. We observed that SM9 males had a higher escape probability than WT males ($p$-value < 0.001, Fig. 5a, Supplementary Table 3), suggesting that the flight capacity of SM9 pseudo-males was at least as high as that of WT male mosquitoes.

We compared SM9 to WT relative fertility (total number of live larvae in a given progeny) and fecundity (total number of eggs laid, estimated by dividing the total number of larvae by the hatching rate). We found no significant difference in SM9 hatching rate ($p$-value = 0.423, Fig. 5b, Supplementary Table 4) or fertility ($p$-value = 0.532, Fig. 5c, Supplementary Table 5) comparing to wild-type control, thus no difference in fecundity either. Then, we measured relative competitiveness between SM9 pseudo-males and wild-type males by mixing equal numbers of transgenic and wild-type males with wild-type females. If males from both lines were equally competitive, we would expect 50% of the females to be inseminated by a WT male (producing WT progeny), and 50% being inseminated by a transgenic pseudo-male. SM9 males producing 55.8% of GFP-positive sons, and their fecundity and fertility being similar to that of WT, we would therefore expect 27.9% of the total progeny from the competition

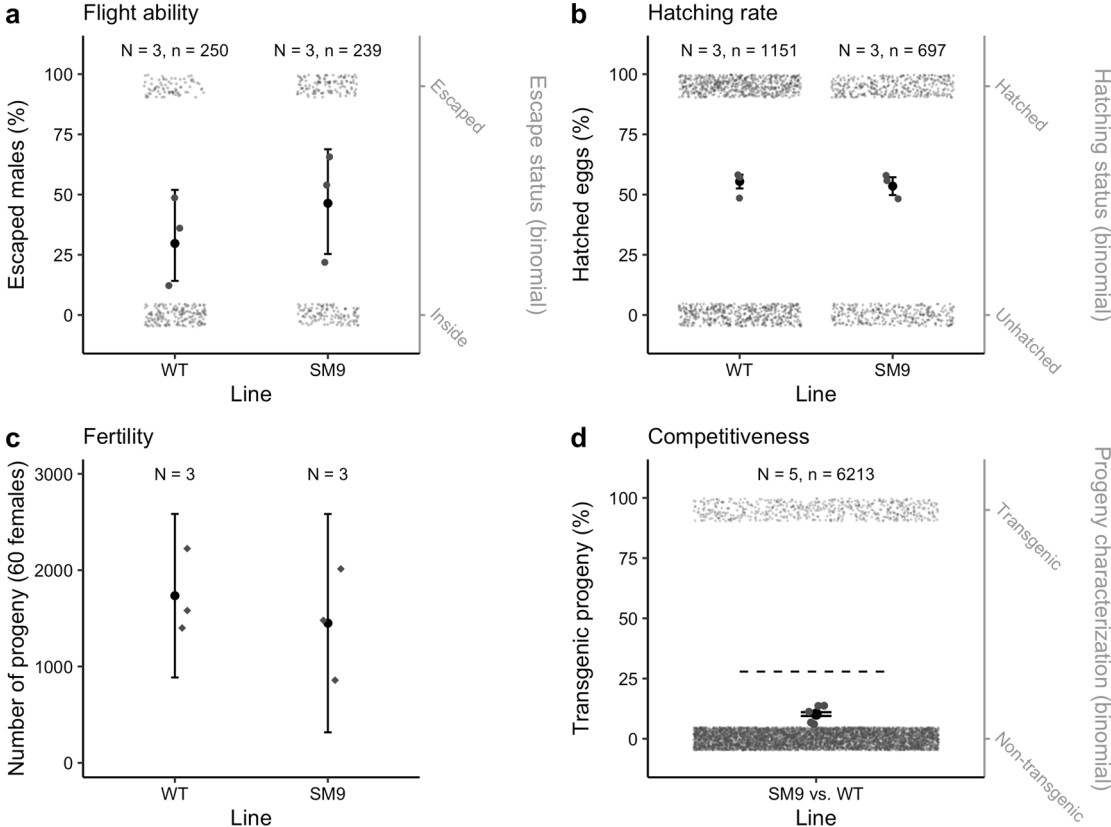

**Fig. 5 Fitness comparison of *Ae. albopictus* SM9 versus WT males. a** Percentage of males that successfully escaped the flight test device. Grey dots in rectangles represent the total numbers of males that remained inside the flight tunnel and that escaped (right *y*-axis). Black dots are the estimate values with vertical lines being 95% confidence intervals. *N* = 3 replicates with an average of 82 ± 13 males were performed. To test the effect of the lines on the flight test success, we used linear generalised mixed-effect model and Bernoulli distribution assumptions with "replicate" as random effect. *p*-value < 0.001. **b** Hatching rate measured by dividing the number of progeny by the number of eggs on *N* = 3 egg batches. Dried eggs were counted, submerged, placed in a vacuum chamber for 30 mn and allowed to hatch for 24 h before counting larvae. Grey dots in rectangles show the total numbers of eggs that hatched or did not hatch (right *y*-axis). Black dots are the estimate values with vertical lines being 95% confidence intervals. Hatching rate was compared by linear generalised mixed-effect model: *p*-value = 0.423. **c** Fertility measured by the number of progeny sired by 30 males crossed with 60 females. Black dots represent the mean value of the *N* = 3 biological replicates, vertical lines represent 95% confidence intervals. The effect of line on fertility was tested using a linear model: *p*-value = 0.532. **d** To estimate SM9 male competitiveness, *N* = 5 competition assays were performed, crossing 30 WT males and 30 SM9 males to 30 females. In their progeny, the percentage of SM9 pseudo-males was measured by COPAS and compared to the expected percentage (dashed line) by linear generalised mixed-effect model: 10.2 ± 0.4 % of transgenic progeny vs. 27.9% of expected value (*p*-value < 0.001). Grey dots in rectangles are representative of the total numbers of transgenic SM9 pseudo-males and non-transgenic individuals in the progeny (right *y*-axis). Black dots are the estimate values with vertical lines being 95% confidence intervals. In all panels, grey dots in the middle part of the graph represent pooled replicates.

assay to be GFP-positive. In this experiment, we observed an estimated mean of 10.2 ± 0.4% of GFP-positive progeny indicating a reduced competitiveness (*p*-value < 0.001, Fig. 5d, Supplementary Table 6). The same competitiveness assay was performed between 1.2G pseudo-males and WT males, and between 3.1 G pseudo-males and WT males (Supplementary Table 6). Both gave a similar result (1.2G pseudo-males vs. WT males 12.5 ± 0.3% of transgenic progeny, *p*-value < 0.001, 3.1G pseudo-males vs. WT males 9.6 ± 0.4% of transgenic progeny, *p*-value < 0.001).

**Automated sex sorting of transgenic pseudo-males**. The transgenesis plasmids carrying a fluorescent marker under strong promoters, neonate larvae can be sorted according to their fluorescence, which, in this case, is sex specific. Similarly to what has been developed in *Anopheles* mosquitoes[31,32], sex sorting can be automated using a COPAS device (Union Biometrica) which allows separation of particles based on fluorescence levels. Using COPAS on the SM9 line, we were able to separate green fluorescent males from non-fluorescent females (Fig. 6a). However, due to the weak activity of the *OpIE2* promoter driving eGFP at

the neonate stage, the fluorescence was not always strong enough to get well separated positive vs. negative clouds by COPAS. Lines 3.1G and 1.2G express higher levels of GFP due to the high activity of the PUb promoter at the neonate stage and provide clearer sex separation (Fig. 6b, c). In all cases, batches of several thousands of neonate larvae could be repeatedly sex separated using COPAS at a speed of approximately 2400 larvae per minute. Visual screening at the pupal stage confirmed perfect sex separation.

## Discussion

The *Nix*-expressing transgenic *Ae. albopictus* lines presented in this work confirm that *Nix* is necessary and sufficient to initiate the male sex determination cascade in *Ae. albopictus*. They also demonstrate that a short version of *Nix* comprising only the first two exons and the first intron is sufficient for complete masculinization. Pseudo-males resulting from the expression of a *Nix* transgene in a female genomic background seem as fecund and fertile as wild-type males.

Interestingly, the genomic location of the *Nix* transgene appears to strongly influence its functionality. While we selected

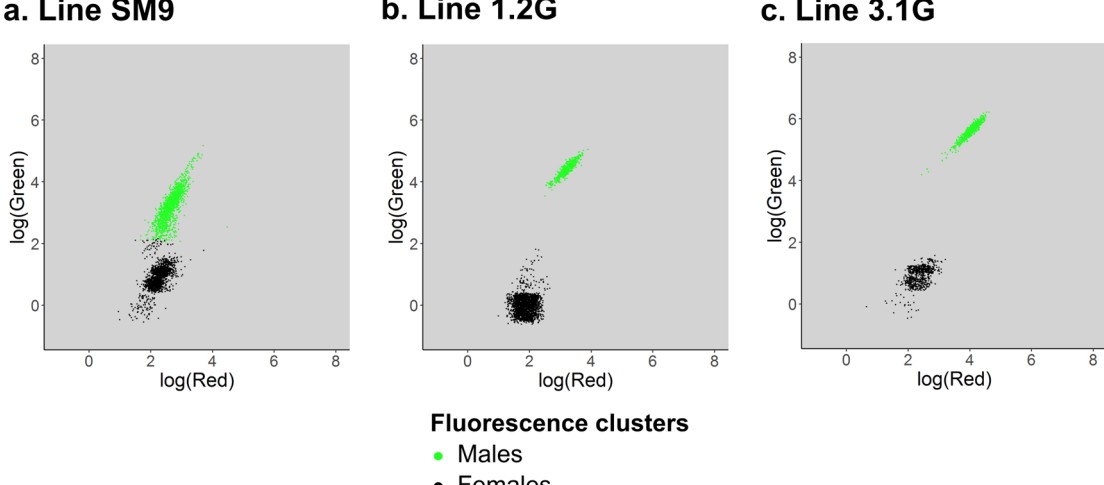

**Fig. 6 Automated sex sorting of *Ae. albopictus* *Nix*-expressing transgenic lines.** Sorting is performed at the neonate stage using a COPAS device. Presented graphs show the fluorescence profile of a representative sample of each line as log(Green) = f(log(Red)). Fluorescence clusters are detected automatically using the 'kmeans' function from the R package 'stat'. *Nix*-expressing pseudo-males being tagged with an *eGFP* marker gene, the top green cluster is composed of males, while females are in the bottom black cluster. **a** 3728 larvae from the SM9 line carrying an *OpIE2-GFP* marker. Here automated clustering detected the lowest male larvae as females, which depicts the difficulty of separating males from females at the neonate stage using an *OpIE2-GFP* marker. **b** 6235 larvae from the 1.2G line carrying a polyubiquitin-*GFP* marker. **c** 1624 larvae from the 3.1G line carrying a polyubiquitin-*GFP* marker.

three independent lines in which expression of the transgene leads to complete conversion of females into functional males, we also discarded many lines that showed either partial masculinisation with deformed genitalia or no masculinisation at all, although their fluorescent transgenesis markers were expressed. Therefore, the activity of the *Nix* promoter is strongly locus-dependent, and full masculinisation requires a certain threshold and/or stage specificity of *Nix* expression. In this work, the random nature of the *piggyBac* insertion greatly facilitated the selection of such amenable loci, whereas it would have been difficult to predict suitable host loci for targeted *Nix* insertion by CRISPR/Cas9. Now that random *piggyBac* insertion has allowed the selection of loci amenable to productive *Nix* transgene expression, an interesting follow-up to this work would be to insert the various *Nix* isoforms in one of these loci. This would allow a more rigorous assessment of whether all isoforms have similar phenotypes. A straightforward way to achieve this would be to excise the lox cassette containing the current transgene via injection of CRE recombinase, to replace it with another isoform by integration into the remaining attP docking site (see Fig. 1).

Similarly to *Ae. aegypti*[24], we observed that pseudo-males harbour molecular characteristics of wild-type males. Of note, we were able to detect an undescribed M-linked homologue of *myo-sex* in the *Ae. albopictus* genome, besides additional autosomal copies, characterized by a 664 nucleotide deletion within the gene's non-coding region. *myo-sex* has been described as responsible for male flight ability in *Ae. aegypti* and is specifically present in wild-type males, hence linked to the M-locus in this species[24]. However, in *Ae. albopictus*, the M-linked copy does not appear to be essential for male flight, as synthetic males expressing a *Nix* transgene but devoid of the M-linked *myo-sex* copy showed similar total *myo-sex* expression level and performed better in flight tests. Moreover, we also identified a strong candidate for a *myo-fem* orthologue, a gene described in *Ae. aegypti* as essential for female flight[29]. In *Ae. albopictus*, we measured that this gene is highly expressed in wild-type females and strongly repressed in wild-type males as well as transgenic pseudo-males.

Interestingly, despite good flight ability, pseudo-males from all three transgenic lines suffered from reduced mating competitiveness. This may result from a negative effect of the ubiquitous expression of the *GFP* marker gene, from disruption of a gene at the transgene insertion sites, or from the absence of unknown factors encoded by the endogenous M-locus involved in male mating ability. Notably, it was observed in *Ae. aegypti* that several long non-coding RNAs of the M-locus were upregulated in the testes and in the male accessory glands[24,33]. We cannot exclude that similar features are present in the *Ae. albopictus* M-locus and that their absence might influence some aspects of the mating process that we were unable to detect in our fertility and fecundity assays.

Besides fully masculinizing *Nix* transgene insertions, we also obtained a number of insertions triggering no or only partial masculinization, whereas the associated fluorescence reporter genes were expressed under control of the polyubiquitin promoter at comparable levels for all obtained lines. This suggests that the genomic context where the *Nix* transgene is inserted affects its expression, and thus, masculinization. There might be a dose-dependency of *Nix* expression during development with a threshold ensuring full masculinization. However, while this might be true for *GFP*-bearing cassettes, we failed to isolate a masculinizing line carrying a *YFP* or a *dsRed* reporter, (i.e., expressing the long *Nix* isoforms 1 or 2). With these isoforms, we achieved partial to full apparent masculinization, but these individuals were apparently infertile. Therefore, it seems that longer isoforms might not be sufficient for masculinization. Alternatively, the short intron retained in our isoform 3–4 construct (marked by eGFP), which we did not include in the longer isoform constructs, could carry a regulatory sequence that is essential for complete masculinization and/or fertility. Finally, it is possible that lines fully masculinized by the longer isoforms would have been recovered had we screened a much larger sample of additional individual transgenic lines.

In transgenic lines where *Nix* transgene expression of specific isoforms is strong enough, *Nix* seems to constitute a self-sufficient ectopic M-locus. Indeed, over >10 generations, the masculinizing lines remained stable, with sex-ratios not differing significantly from the wild-type strain. This result is consistent with the high flexibility of sex determination in Diptera. This flexibility is

probably best illustrated in *Musca domestica* in which the *Mdmd* male factor is mobile between chromosomes via translocation events[34–36]. Even though it is believed that the *Aedes* non-recombining sex-loci are currently evolving towards sex-chromosomes[37], this experiment shows that incipient chromosome differentiation is not essential to the species' viability. Interestingly, the observed antero-posterior gynandromorphism after injection of a plasmid expressing CRE-recombinase into SM9 embryos shows that sex determination is tissue-autonomous in *Ae. albopictus* and is entirely consistent with the cell-autonomous function of *dsx* observed in *D. melanogaster*[38–40].

Finally, from an operational point of view, the masculinizing *Nix*-expressing transgenic lines could be used as genetic sexing strains, since their fluorescence genes are tightly linked to the male-determining gene. Despite SM9 pseudo-males performing well in flight assays, their competitiveness in mating assays confronting equal numbers of wild-type and pseudo-males the progeny sired by pseudo-males was 2.7 fold reduced compared to the expected 50%. With proper adjustment in the number of males to be released in a SIT intervention[41,42], this line still has strong potential for field deployment. One may also be interested in measuring the competitiveness of additional, newly generated masculinizing lines since the transgene insertion may have disrupted an important gene in the lines we tested.

In conclusion, in *Aedes albopictus*, *Nix* is confirmed to be necessary and sufficient for complete masculinization. The developed *Nix*-expressing m/m lines carrying a reporter fluorescence transgene are a promising starting point for the development of highly stable genetic sexing strains.

## Methods

**Plasmid construction.** Cloned genomic sequences were either amplified by PCR using Phusion™ High-Fidelity DNA Polymerase (ThermoFisher Scientific, France), or ordered as gBlocks from IDT DNA (Belgium). Cloning of PCR products and gBlocks in the pKSB-intermediate vector[43] was performed using NEBuilder® HiFi DNA Assembly (New England Biolabs, France) following the manufacturer's instructions. Modules were then assembled by Golden Gate Cloning into a destination *piggyBac* vector backbone (Addgene #173496) using the Eco31I restriction enzyme (Thermo Scientific, ThermoFisher Scientific, France). Plasmids were purified using NucleoSpin Plasmid and NucleoBond Xtra Midi EF kits (Macherey Nagel, France) according to the manufacturer's instructions. Sequence verifications were performed using Eurofins Genomics (Germany) Sanger sequencing services.

Four *Nix*-expressing piggyBac-based transgenesis plasmids were built, all made available along with their annotated nucleotide sequence at Addgene (#173505, #173665, #173666, #173667). The *Nix* endogenous promoter was captured by PCR amplification of a 2kbp region upstream of the first exon (Supplementary Note 1). A plasmid composed of the *Nix* endogenous promoter plus the four exons, together with an *Ae. aegypti* polyubiquitin (AePUb)-*dsRed* reporter gene, was built to express and track isoform 1 according to the latest terminology[22]. A plasmid composed of the *Nix* endogenous promoter plus exons 1, 3 and 4 and an AePUb-YFP reporter gene was built to express and track isoform 2. Two plasmids encoding the *Nix* endogenous promoter, the first exon, the first intron and the second exon and the eGFP reporter gene under the control of either the *OpIE2* promoter or the AePUb promoter were both used to express and track isoforms 3 and 4.

**Embryo microinjections.** Embryo microinjections were performed as previously described for *Anopheles* eggs using a Nikon Eclipse TE2000-S inverted microscope, an Eppendorf Femtojet injector and a TransferMan NK2 micromanipulator[43], with injection mixes comprising 320 ng/μl of *piggyBac* construct and 80 ng/μl of transposase-expressing helper plasmid. We codon-optimized the sequence of *piggyBac* transposase for mosquitoes, added an extra nuclear localization signal to the transposase sequence, and placed it under the control of the *Aedes aegypti* PUb promoter. We observed these changes to result in a massive increase in the integration efficiency of the *piggyBac* construct. The plasmid and its sequence are available upon request. For injections with CRE recombinase, the injected solution also contained a neutral plasmid expressing *DsRed* gene under the control of the PUb promoter to control for microinjection efficiency. Surviving larvae expressing eGFP and transiently showing red fluorescence were collected and examined again at the pupal stage. All larvae counting and fluorescence sorting were performed using a COPAS SELECT device (Union Biometrica, Belgium) with the provided Biosort software.

**Mosquito rearing.** *Aedes albopictus* mosquitoes (strain BiA) were collected as larvae from a garden rainwater collector in the city of Bischheim, near Strasbourg (France) in 2018 and maintained in the insectary since then at 25 °C, 75–80% humidity, with a 14-h/10-h light/dark photoperiod. Larvae were reared in pans filled with demineralized water and provided ground TetraMin fish food twice a day. Adult mosquitoes were caged and provided with 10% sugar solution *ad libitum*. Females were blood-fed on anesthetised mice. Eggs were laid three days later on wet kraft paper and allowed to develop for another three days before being dried.

**Molecular characterization of mosquito lines.** Genomic DNA was extracted from pupae using NucleoSpin Tissue kit following the manufacturer's instructions (Macherey Nagel, France). Total RNA was extracted using TRIzol RNA Isolation Reagents (Invitrogen, ThermoFisher Scientific, France) and reverse-transcribed using RevertAid H Minus First Strand cDNA Synthesis Kit (Thermo Scientific, ThermoFisher Scientific, France). Presence of endogenous and/or exogenous *Nix* in the genome was assessed by PCR using GoTaq® Green Master Mix (M712, Promega, France) according to the manufacturer's recommendations with the Nix-833 published primers[25] and primer EM1926-EM1927 (Supplementary Table 1). The *dsx* and *fru* splicing patterns were assessed by RT-PCR using published primers[44] with Phire Tissue Direct PCR Master Mix (Thermo Scientific, ThermoFisher Scientific, France). All PCRs were performed in a Veriti 96-wells Thermal Cycler (Applied Biosystems, ThermoFisher Scientific, France). Relative expression of *Nix*, *myo-sex*, and *myo-fem* orthologues was assessed by RT-qPCR with published primers[22] as well as primers provided in Supplementary Table 1. All measurements were performed in triplicates. The *AalRpS7* housekeeping gene was used as an endogenous reference with published primers[44] for normalization of the expression level using the $\Delta C_T$ method. qPCRs were performed using SYBR™ Fast SYBR™ Green Master Mix (ThermoFisher Scientific, France) and a 7500 Fast Real-Time PCR System machine (Applied Biosystems, ThermoFisher Scientific, France).

**Phenotypical characterization of mosquito lines.** For all phenotypical tests, mosquitoes from the reference WT strain (BiA) and from the tested transgenic strain were hatched and reared mixed together, with the same larval density and the same amount of food in all tanks. The hatching rate was estimated by counting eggs on a piece of egg paper under a binocular microscope. Eggs were then submerged, placed in a vacuum chamber at 25% of atmospheric pressure for 30 min and allowed to hatch for 24 h before counting the number of larvae by COPAS, on $N = 3$ separate egg batches. Sex ratio was measured on $N = 3$ samples of each line. WT males and females were counted manually at the pupal stage based on genitalia observation under a binocular microscope[45]. The number of males and females in the transgenic lines was counted at the neonate stage using a COPAS based on fluorescence. Right wings from $N = 38$ WT males, $N = 39$ WT females and $N = 48$ SM9 males were dissected under a binocular microscope and placed on a microscope slide using double-sided tape. Pictures were taken under a Zeiss SteREO binocular microscope with X-Cite Xylis engine (Excelitas technologies) and analysed using ImageJ software[46] to measure wing length as a proxy for body size[27]. Male flight ability was assessed using a Flight Test Device as previously described, with N = 3 replicates of 82 ± 13 males for each treatment[30]. Relative fertility was calculated by comparing the number of live larvae in the progeny from 30 SM9 pseudo-males crossed with 60 wild-type females vs. 30 wild-type males crossed with 60 wild-type females. $N = 3$ independent replicates of this test were performed for each strain. Relative competitiveness was estimated by placing 30 pseudo-males of a *Nix*-expressing line with 30 males of the wild-type line in a cage with 30 wild-type females in $N = 5$ independent replicates. In their progeny, we measured the percentage of *Nix*-expressing pseudo-males (tracked by fluorescence using a COPAS device). If fecundity and fertility are similar in both lines, and if they were equally competitive, half the females would mate with a WT male and the other half with a transgenic pseudo-male, meaning that 50% of the progeny would have been sired by transgenic pseudo-males. Within the offspring of the pseudo-transgenic males, the percentage of males would only depend on the line's sex ratio. Consequently, the expected percentage of transgenic males (fluorescent larvae) in the total progeny of the competition assay would equal half the percentage of males in the transgenic strain. This theoretical value is then compared to the measured percentage of transgenic males.

**Animal care.** Animal care was in accordance with CNRS guidelines and approved by the CREMEAS ethics committee.

**Statistics and reproducibility.** '±' symbols in main text represent the data standard deviation (SD) or model estimation standard errors (SE). Error bars in Figs. 3, 4 and 5 are 95% confidence intervals (CI). The number of biological replicates (*N*), total number of individuals (*n*) and type of statistical analysis are indicated in methods and figure legends. For qPCR, three technical replicates of each biological sample were used and each line was tested on three independent biological samples. To test the effects of the lines on gene expression we used linear model and normal distribution assumptions. The significance between the lines was tested by ANOVA followed by pairwise Tukey test. The effect of lines on wing length and fertility were tested using linear model and normal distribution assumptions. The effect of the

lines on flight ability, hatching rates and sex ratios were tested using linear generalised mixed-effect model and binomial distribution assumptions. The replicate was set as random effect for sex ratios and flight tests as experiments were performed on different days. The relative competitiveness of transgenic lines compared to WT was tested using linear generalised mixed-effect model and binomial distribution assumptions. For this test, we compared the number of SM9 male progeny measured in the offspring of each competitiveness replicate, to the expected number of progeny that would have been obtained if both lines were as competitive. All statistical analyses and plots were performed on R software version 4.0.5[47]. All model assumptions were tested using the R package 'performance' (Supplementary Data 1).

**Reporting summary**. Further information on research design is available in the Nature Research Reporting Summary linked to this article.

## Data availability

Plasmid sequences have been deposited on Addgene (https://www.addgene.org/) under reference numbers: #173505, #173665, #173666, #173667 for *Nix*-expressing *piggyBac*-based transgenesis plasmids and #173496 for *piggyBac* vector backbone. All plasmids and mosquito strains described in this paper are available upon request to E.M. All data needed to evaluate the conclusions in the paper are present in the paper and/or the Supplementary Information.

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

## Acknowledgements

This study was funded by EU ERC CoG—682387 REVOLINC to J.B. The contents of this publication are the sole responsibility of the authors and do not necessarily reflect the views of the European Commission. Mosquito production and insectarium operation were supported by Agence Nationale de la Recherche grant #ANR-11-EQPX-0022. Part of the technical work was funded by ANR grants #ANR-19-CE35-0007 GDaMO and # 18-CE35-0003-02 BAKOUMBA to E.M.

## Author contributions

C.L., T.B., J.B. and E.M. designed research. C.L., R.P.O. and E.M. performed research and analysed data. C.L. wrote the paper with inputs from all authors.

## Competing interests

The authors declare no competing interests.

## Additional information



