## [Peer Review File · Communications Biology]

Reviewers' comments:

Reviewer #1 (Remarks to the Author):

The manuscript by Lutrat et al., describes the development of a number of transgenic *Aedes albopictus* lines that express a Nix transgene together with a fluorescent marker. After appropriate crosses, one of the lines, SM9 consisted of only genetic females, around half of which were fluorescently marked and phenotypically masculinized (pseudomales). These pseudo-males, unlike those of *Ae. aegypti*, are fully able to fly, apparently due to the presence of at least one non M-linked myo-sex gene. They also show similar fertility and fecundity compared to wild-type males. Experimental excision of the transgenic cassette resulted in demasculinization and loss of fluorescence near the site of injection.

The manuscript describes a series of very interesting experiments and significantly extends the work performed on the Nix locus in *Ae. albopictus*. It shows that Nix is necessary and sufficient for masculinization and that it is possible to develop genetic sexing strains using appropriate Nix transgenes.

The experimental procedures, analyses and conclusions are appropriate. I have only a few minor comments on the manuscript:

Line 44: to suppress mosquito populations,

Line 66: high similarity to *Ae. aegypti* Nix

Lines 83 and 100: In Figure 1 (and in the text) give some indication as to which plasmid is being referred to (perhaps by labelling the schematic representations a - d or similar).

Line 115-119: Isn't there a third possibility -that the fluorescent males are a mixture of masculinised females and natural males (both carrying one or more fluorescent transgenes)?

Line 129: what are negative females? wild-type or GFP-?

Line 225-227: The logic behind this phrase is not sufficiently clear (presumably matings between SM9 males and WT females give rise to 50% fluorescent male larvae and non-fluorescent female larvae) - please clarify.

Line 351: The massive increase in efficiency was observed by the authors or has been published elsewhere (reference).

Line 391: Eggs were then submerged, placed in a ...

Line 392: 30min

Line 395: perhaps include a reference for sexing pupae (such as Moorefield, H.H. 1951 Sexual dimorphism in mosquito pupae. *Mosquito News*, 11:3)

Line 406: Nix-expressing

Figure 3a: Do the rectangles containing dots for each category (male/female) facilitate understanding the data. I would suggest removing the two categories on the right axis (and the rectangles with dots) and leave the % males on the left axis and the N and n values for each line at the top. The same suggestion for Figure 5a,b & d.

Reviewer #2 (Remarks to the Author):

This is an interesting paper and, while imperfect in the consistency of male conversion across all the transgenic lines, it does provide strong data showing that nix acts as expected in *Aedes albopictus* and that this, in conjunction with a genetic sexing maker, can lead to efficient genetic sexing of males and females in the converted lines.

The strengths of the paper are:

1. demonstration that nix can result in complete conversion to male in many strains
2. the use of this in *Aedes albopictus* which is a serious pest of pathogens but in which little transgenesis-based work has been done relative to *Aedes aegypti*.
3. demonstration that these genetic manipulations can result in efficient, automated genetic sexing which is a decades-old goal of many approaches to bringing molecular biological tools to pest

insects.

4. The quality of the analysis and the experimental approach.

The weakness are:

1. The use of transposon-mediated transgenesis rather than CRISPR-mediated transgenesis. However I will concede the point that the later may be difficult at the moment in this species, especially with HDR-mediated integration being required.
2. As the authors conceded, the use of PB and its random insertion into the genome. may explain some of their negative results. Once again this is a limitation of *Albopictus* in which techniques such as RCME have not, to my knowledge, been applied.
3. the slight reduction in genetic fitness.

On balance the positives out weight the negatives and I think this work will receive much attention.

I recommend it be accepted.

Answers to reviewers' comments

We are grateful to both reviewers for their appreciation of our work and for these helpful and constructive comments. Please find below point-by-point answers to the questions raised by the reviewers plus a series of modifications we made to the manuscript.

Reviewer #1 (Remarks to the Author):

General comments:

“The manuscript by Lutrat et al., describes the development of a number of transgenic *Aedes albopictus* lines that express a Nix transgene together with a fluorescent marker. After appropriate crosses, one of the lines, SM9 consisted of only genetic females, around half of which were fluorescently marked and phenotypically masculinized (pseudomales). These pseudo-males, unlike those of *Ae. aegypti*, are fully able to fly, apparently due to the presence of at least one non M-linked myo-sex gene. They also show similar fertility and fecundity compared to wild-type males. Experimental excision of the transgenic cassette resulted in demasculinization and loss of fluorescence near the site of injection.

The manuscript describes a series of very interesting experiments and significantly extends the work performed on the Nix locus in *Ae. albopictus*. It shows that Nix is necessary and sufficient for masculinization and that it is possible to develop genetic sexing strains using appropriate Nix transgenes.

The experimental procedures, analyses and conclusions are appropriate. I have only a few minor comments on the manuscript:”

Answers to comments

Nb	Comment	Answer
1	Line 44: to suppress mosquito populations	Edited (line 41)
2	Line 66: high similarity to Ae. aegypti Nix	Edited (line 63)
3	- Lines 83 and 100: In Figure 1 (and in the text) give some indication as to which plasmid is being referred to (perhaps by labelling the schematic representations a - d or similar).	Edited in the text (lines 80 and 96-97) and on the figure, thank you for the suggestion. We also noticed some small mistakes on our figure, which are now fixed.
4	Line 115-119: Isn't there a third possibility -that the fluorescent males are a mixture of masculinised females and natural males (both carrying one or more fluorescent transgenes)?	Indeed, we did observe this in the first generations of some lines. We eliminated this complexity by generating single-male families, crossing individual transgenic males with WT females as mentioned in lines 105-109 until we could detect only one fluorescent marker and a sex ratio close to 50%. To further exclude the possible co-existence of M-linked and autosomal insertions of the same marker, we PCR-screened several individual males of each single-male line to verify that the presence/absence of an endogenous M-locus was the same in all these males (shown in panel b of Supplementary Figure 2 for SM9 line for example).

		Following this step, what could persist are multiple insertions of the same type (either autosomal or M-linked) and of the same fluorescence. We did observe this in some of the lines carrying autosomal masculinizing insertions that still produced fluorescent females or intersex individuals. Hence, we backcrossed individual males for a few more generations in the lines of interest until we got rid of any other non-fully masculinizing insertions (lines 127-131).
5	Line 129: what are negative females? wild-type or GFP-?	They are wild-type, it has been edited for more clarity (line 128).
6	Line 225-227: The logic behind this phrase is not sufficiently clear (presumably matings between SM9 males and WT females give rise to 50% fluorescent male larvae and non-fluorescent female larvae) - please clarify.	This part has been rephrased and developed so that the logic is easier to understand (lines 225-231).
7	Line 351: The massive increase in efficiency was observed by the authors or has been published elsewhere (reference).	It is an observation from the authors (made clearer in line 368).
8	Line 391: Eggs were then submerged, placed in a ...	Edited (line 409)
9	Line 392: 30min	Edited (line 410)
10	Line 395: perhaps include a reference for sexing pupae (such as Moorefield, H.H. 1951 Sexual dimorphism in mosquito pupae. Mosquito News, 11:3)	Thank you for the suggestion, we included it (line 413).
11	Line 406: Nix-expressing	Edited (line 425)
12	Figure 3a: Do the rectangles containing dots for each category (male/female) facilitate understanding the data. I would suggest removing the two categories on the right axis (and the rectangles with dots) and leave the % males on the left axis and the N and n values for each line at the top. The same suggestion for Figure 5a,b & d.	These data have been analysed using a model taking into account each mosquito as an individual replicate with a binomial status (male versus female, transgenic versus non-transgenic etc.). These individual replicates are what is being represented by dots in these rectangles. We believe that showing them is, in this case, as relevant as showing the pooled replicates because of the statistical analyses we performed (see Suppl. Data 1). We have modified the figures so that the contrast between what is referred to on the left and right axes is stronger, and hopefully, easier to understand.

Reviewer #2 (Remarks to the Author):

General comments

“This is an interesting paper and, while imperfect in the consistency of male conversion across all the transgenic lines, it does provide strong data showing that *nix* acts as expected in *Aedes albopictus* and that this, in conjunction with a genetic sexing maker, can lead to efficient genetic sexing of males and females in the converted lines.

The strengths of the paper are:

1. demonstration that *nix* can result in complete conversion to male in many strains
2. the use of this in *Aedes albopictus* which is a serious pest of pathogens but in which little transgenesis-based work has been done relative to *Aedes aegypti*.
3. demonstration that these genetic manipulations can result in efficient, automated genetic sexing which is a decades-old goal of many approaches to bringing molecular biological tools to pest insects.
4. The quality of the analysis and the experimental approach.

The weakness are:

1. The use of transposon-mediated transgenesis rather than CRISPR-mediated transgenesis. However I will concede the point that the later may be difficult at the moment in this species, especially with HDR-mediated integration being required.
2. As the authors conceded, the use of PB and its random insertion into the genome. may explain some of their negative results. Once again this is a limitation of *Albopictus* in which techniques such as RCME have not, to my knowledge, been applied.
3. the slight reduction in genetic fitness.

On balance the positives out weight the negatives and I think this work will receive much attention.

I recommend it be accepted.”

Answers on the study's weaknesses

Nb	Comment	Answer
1	The use of transposon-mediated transgenesis rather than CRISPR-mediated transgenesis. However I will concede the point that the later may be difficult at the moment in this species, especially with HDR-mediated integration being required.	Indeed our attempts to target the Nix locus using CRISPR/Cas9 were unsuccessful, though we were able to obtain CRISPR knock-ins in other loci of the Aedes albopictus genome. However, the main reason why we favoured piggyBac over CRISPR in this experiment lies in the ability of Nix transgenes to trigger masculinisation, which is highly dependent on the genomic location where the transgene lands, as shown by our results (some insertions failed to masculinize females). Thus, the random nature of piggyBac insertion was instrumental to select loci allowing full functionality of the Nix transgenes. We have made this clearer in the Discussion (lines 260-268)
2	As the authors conceded, the use of PB and its random insertion into the genome. may explain some of their negative results. Once again this is a limitation of Albopictus in which techniques such as RCME have not, to my knowledge, been applied.	We agree that docking site transgenesis would have allowed us to compare more rigorously the masculinising effect of the different Nix isoforms. This is actually an interesting perspective opened by our work, as the lox cassette and attP docking site in our constructs can now be used to exchange

		one isoform for another. We thank the reviewer for this idea and add it in the discussion as a perspective (lines 268-274).
3	the slight reduction in genetic fitness	Three hypotheses to explain the fitness reduction are proposed in the discussion : (i) a negative effect of the ubiquitous expression of the GFP marker gene, (ii) disruption of a gene at the transgene insertion sites, and (iii) the absence of unknown factors encoded by the endogenous M-locus involved in male mating ability (lines 287-291). The 1 st and 3 rd hypotheses seem more likely than the 2 nd one as the competitiveness was reduced in three independent lines with distinct integration sites. Current state of our knowledge did not allow us to test them. However, the reduced genetic fitness only affects the potential application as a sex sorting strain, which was not the 1 st objective of this study, and which remains interesting given the great cuttings they allow in male rearing cost.

Other modifications made to the manuscript:

Main text

Line 14: Replaced “These authors contributed equally to this work” with “Jointly supervising authors” as suggested in the authors’ guidelines.

Line 127: Added “(hereafter termed pseudo-males)”

Line 130: Added “pseudo-”

Methods

Line 369 and line 460: Addgene deposition of the piggyBac transposase helper plasmid has been denied because of pre-existence of a patent on piggyBac transposase. We changed its Addgene initial identifier by a statement of availability upon request.

Line 410: Added “at 25% of atmospheric pressure”

Line 437: Renamed the “Statistical analysis” part in “Statistics and Reproducibility” accordingly to the journal formatting.

Figures

Figure 1: Schematic representation of *Ae. albopictus* Nix isoform cloning in the injected plasmids. Isoform naming follows work published in ²². Grey boxes are Nix exons, with darker grey being the translated parts. White triangles are piggyBac 5' and 3' inverted terminal repeat (ITR) sequences, orange boxes represent an AttP landing site included for potential future purposes, purple boxes represent loxP recombination sites, white arrows are promoters, pink polygons are SV40 polyA sequences, green, red and yellow boxes are eGFP, DsRed and YFP gene sequences, respectively. Drawing not to scale. Plasmids carry **a)** Nix isoforms 3-4 and a GFP marker gene under the control of the OpIE2 promoter, **b)** Nix isoforms 3-4 and a GFP marker gene under the control of the polyubiquitin promoter, **c)** Nix isoform 1 and a DsRed marker gene under the control of the polyubiquitin promoter, and **d)** Nix isoform 2 and a YFP marker gene under the control of the polyubiquitin promoter. Detailed plasmid sequences can be found under Addgene references #173505, #173665, #173666, #173667.

Changes to Figure 1:

- Added a, b, c, d letters and legends.
- Corrected DsRed and YFP positions with the correct isoform
- Corrected start and stop codons on Nix isoforms

Figure 2: Tissue demasculinization upon CRE/lox excision of the transgenic *Nix* cassette. **a)** Representative transgenic male pupa and male adult from the SM9 line. **b)** Representative non-transgenic female pupa and female adult from the SM9 line. **c)** Transgenic SM9 male pupa and adult injected as embryos in the posterior pole with CRE-recombinase that excised the *Nix*-eGFP cassette in the injected region. These individuals show a male anterior body with female genitalia. **d)** Transgenic SM9 male pupa and adult injected with CRE-recombinase in the anterior pole of the embryo. Note the female anterior body and male genitalia.

Changes to Figure 2:

- Same image exported in higher quality

Figure 3: Comparison of sex ratios and wing lengths between transgenic and WT lines. a) Sex ratio comparison between the WT line and the SM9, 1.2G and 3.1G transgenic lines. Sex ratio of the WT line was counted manually on N=3 independent batches of pupae, while sex ratios of transgenic lines were counted on N=3 independent batches of neonate larvae using COPAS. **Grey dots in rectangles** represent the total numbers of males and females (right y-axis). Black dots are the estimate values with vertical lines being 95% confidence intervals. Sex ratios were compared using linear generalised mixed-effect model. None of the sex ratios were significantly different from that of the WT line: SM9 vs. WT p -value = 0.165, 1.2G vs. WT p -value = 0.528, 3.1G vs. WT p -value = 0.760. **b)** Wing length comparison between N=38 wild-type males, N=39 wild-type females and N=48 *Nix*-expressing SM9 pseudo-males represented as violin plots with jitter grey data points. Wing length was measured on ImageJ software from pictures of dissected right wings taken under a binocular microscope. Comparisons were performed using linear model: WT male vs. WT female p -value < 0.001, SM9 male vs. WT female p -value < 0.001, WT male vs. SM9 male p -value = 0.998.

Changes to Figure 3:

- Made the right y-axis on panel a) light grey to make clearer that it corresponds to the top and bottom dots.

Figure 4: Relative expression of *Nix*, *myo-sex* and *myo-fem* orthologue in *Ae. albopictus* transgenic males. RT-qPCR results are represented by $-\Delta C_T$, which reflects the relative expression level of each gene in a given treatment, C_T values being inversely proportional to the expression levels. *AalRps7* was used as endogenous reference gene. Grey dots represent each data point. Black dots represent the mean value of the $N = 3$ biological replicates, vertical lines represent 95% confidence intervals. On each panel, distinct letters represent significant difference in an ANOVA followed by a pairwise Tukey test (p -value < 0.001). **a)** *Nix* relative expression. **b)** Relative comparison of *myo-sex* orthologues total expression levels. **c)** Relative expression of the candidate orthologue of the *Ae. aegypti myo-fem* gene, LOC109402113. **d)** Relative expression of the candidate orthologue LOC115254984, which is annotated as a putative pseudo-gene. This primer pair could also amplify LOC115254986 due to high sequence similarity but this other pseudo-gene seems not to be expressed.

Changes to Figure 4:

- Spread replicate dots around the confidence intervals in order to make them more visible.

Figure 5: Fitness comparison of *Ae. albopictus* SM9 versus WT males. **a)** Percentage of males that successfully escaped the flight test device. Grey dots are representative of the total numbers of males that remained inside the flight tunnel and that escaped (right y-axis). Black dots are the estimate values with vertical lines being 95% confidence intervals. N = 3 replicates with an average of 82 ± 13 males were performed. To test the effect of the lines on the flight test success, we used linear generalised mixed-effect model and Bernoulli distribution assumptions with “replicate” as random effect. p -value < 0.001. **b)** Hatching rate measured by dividing the number of progeny by the number of eggs on N = 3 egg batches. Dried eggs were counted, submerged, placed in a vacuum chamber for 30mn and allowed to hatch for 24h before counting larvae. **Grey dots in rectangles represent** the total numbers of eggs that hatched or did not hatch (right y-axis). Black dots are the estimate values with vertical lines being 95% confidence intervals. Hatching rate was compared by linear generalised mixed-effect model: p -value = 0.423. **c)** Fertility measured by the number of progeny sired by 30 males crossed with 60 females. Grey dots represent each data point. Black dots represent the mean value of the N = 3 biological replicates, vertical lines represent 95% confidence intervals. The effect of line on fertility was tested using a linear model: p -value = 0.532. **d)** To estimate SM9 male competitiveness, N = 5 competition assays were performed, crossing 30 WT males and 30 SM9 males to 30 females. In their progeny, the percentage of SM9 pseudo-males was measured by COPAS and compared to the expected percentage (dashed line) by linear generalised mixed-effect model: 10.2 ± 0.4 % of transgenic progeny vs. 27.9% of expected value (p -value < 0.001). **Grey dots show** the total numbers of transgenic SM9 pseudo-males and non-transgenic individuals in the progeny (right y-axis). Black dots are the estimate values with vertical lines being 95% confidence intervals.

Changes to Figure 5:

- Made the right y-axes on panels a) b) and d) light grey so that it is clearer they correspond to the top and bottom dots.

- Made the dots lighter as well

Figure 6: Automated sex sorting of *Ae. albopictus* *Nix*-expressing transgenic lines. Sorting is performed at the neonate stage using a COPAS device. Presented graphs show the fluorescence profile of a representative sample of each line as $\log(\text{Green}) = f(\log(\text{Red}))$. Fluorescence clusters are detected automatically using the 'kmeans' function from the R package 'stat'. *Nix*-expressing pseudo-males being tagged with an eGFP marker gene, the top green cluster is composed of males, while females are in the bottom black cluster. **a)** 3,728 larvae from the SM9 line carrying an OpIE2-GFP marker. Here automated clustering detected the lowest male larvae as females, which depicts the difficulty of separating males from females at the neonate stage using an OpIE2-GFP marker **b)** 6,235 larvae from the 1.2G line carrying a polyubiquitin-GFP marker. **c)** 1,624 larvae from the 3.1G line carrying a polyubiquitin-GFP marker.

Changes to Figure 6:

- Replaced the COPAS screen images by plots obtained using COPAS txt cytometry data (different samples of the same line) for higher image quality. Plots done on R with automated cluster detection.

Supplementary Information

All source data have been included in Supplementary Information.

REVIEWERS' COMMENTS:

Reviewer #1 (Remarks to the Author):

I am happy with the authors' replies to my comments and those of the other reviewer. I recommend the manuscript for publication.